# The Absence of Hydrodynamic Stress Promotes Acquisition of Freezing Tolerance and Freeze-Dependent Asexual Reproduction in the Red Alga ‘*Bangia*’ sp. ESS1

**DOI:** 10.3390/plants10030465

**Published:** 2021-03-01

**Authors:** Yoshiki Omuro, Ho Viet Khoa, Koji Mikami

**Affiliations:** 1Department of Aquaculture Life Science, School of Fisheries Sciences, Hokkaido University, 3-1-1 Minato-cho, Hakodate 041-8611, Japan; rattpa68@eis.hokudai.ac.jp; 2Division of Marine Life Science, Graduate School of Fisheries Sciences, Hokkaido University, 3-1-1 Minato-cho, Hakodate 041-8611, Japan; hvkhoa59@gmail.com; 3Department of Food Recourse Development, School of Food Industrial Sciences, Miyagi University, 2-2-1 Hatatate, Taihaku-ku, Sendai 982-0215, Japan

**Keywords:** asexual reproduction, ‘*Bangia*’ sp. ESS1, Bangiales, calm stress, freezing tolerance, fatty acid, membrane fluidity

## Abstract

The ebb tide causes calm stress to intertidal seaweeds in tide pools; however, little is known about their physiological responses to loss of water movement. This study investigated the effects of static culture of ‘*Bangia*’ sp. ESS1 at 15 °C on tolerance to temperature fluctuation. The freezing of aer-obically cultured thalli at −80 °C for 10 min resulted in the death of most cells. By contrast, statically cultured thalli acquired freezing tolerance that increased cell viability after freeze–thaw cycles, although they did not achieve thermotolerance that would enable survival at the lethal temperature of 32 °C. Consistently, the unsaturation of membrane fatty acids occurred in static culture. Notably, static culture of thalli enhanced the release of asexual spores after freeze-and-thaw treatment. We conclude that calm stress triggers both the acquisition of freezing tolerance and the promotion of freezing-dependent asexual reproduction. These findings provide novel insights into stress toler-ance and the regulation of asexual reproduction in Bangiales.

## 1. Introduction

Bangiales is an order of red algae characterized by thalli with a filamentous or leafy shape [1,2]. These seaweeds are sessile multicellular organisms that live in intertidal regions, where temperature, salinity, and nutritional conditions usually fluctuate. Recent physiological and “omics” analyses indicate that Bangiales respond to heat, cold, salinity, hyper-osmolality, and desiccation through stress-inducible gene expression and repression [3,4,5,6,7,8,9]. Thus, Bangiales sense environmental changes as different abiotic stresses and express or repress different sets of genes for each stress. Since thalli of Bangiales mostly appear in winter and early spring, acclimation to low-temperature stress seems to be essential for their growth and survival; however, little is known about how Bangiales acquire tolerance to cold stress.

Cold acclimation is a phenomenon in which the exposure of plants to non-freezing (chilling) temperature promotes the acquisition of tolerance to freezing at sub-zero temperature [10,11,12,13,14]. Cold acclimation has been observed in both micro- and macroalgae [15,16,17]. In terrestrial plants, cold acclimation is established via exposure to other stresses [18,19,20]. For instance, desiccation stress induced freezing tolerance in winter cereals [21,22,23]. A similar phenomenon has been observed in microalgae [17]. These findings suggest that the ability to acquire freezing tolerance by exposure to environmental stresses other than low temperature might be conserved among photosynthetic organisms. However, it has not yet been examined whether cold acclimation is established by such a cross-tolerance mechanism in macroalgae.

There is a close relationship between an increase in cold-stress tolerance and membrane fluidization via the unsaturation of membrane fatty acids in poikilothermic organisms [24,25,26], which can be demonstrated by the artificial unsaturation of membrane fatty acids via genetic transformation using fatty acid desaturase genes [27,28,29,30,31,32,33,34]. Recently, cold stress-induced unsaturation of membrane fatty acids was also reported in the red seaweed *Bangia fuscopurpurea* [5]. Thus, the monitoring of changes in the membrane fatty acid composition can serve as a powerful tool to evaluate physiological responses related to freezing tolerance via cold acclimation in algae.

The ebb tide and resulting loss of water flow—the most drastic change in living conditions at the intertidal region—exposes Bangiales to temperature changes, desiccation, nutritional starvation, and other potential stresses. We hypothesized that loss of water movement might strengthen the effects of environmental changes on growth and viability and thus trigger the acquisition of tolerance to abiotic stresses in Bangiales. The filamentous red seaweed ‘*Bangia*’ sp. ESS1 [35] is used as a model organism to investigate the stress responses of Bangiales in our laboratory. Using this species, we previously reported an acceleration of asexual reproduction under heat-stress conditions [36] and confirmed that heat-stress memory has an intrinsic ability to induce thermotolerance [37]. We also established a transient gene expression system [38] and identified reference genes to quantify gene expression under various kinds of abiotic stress [35]. Therefore, to test our hypothesis, we employed ‘*Bangia*’ sp. ESS1 and focused on loss of water movement due to ebb tide as an abiotic stress. We investigated the effects of static culture at 15 °C, a regular laboratory culture temperature, on the acquisition of tolerance to temperature fluctuation and on membrane fatty acid compositions.

## 2. Results

### 2.1. Acquisition of Freezing Tolerance by Exposure to Calm Stress

When thalli of ‘*Bangia*’ sp. ESS1 were grown under hydrodynamic stress by aeration culture at 15 °C, frozen at −80 °C for 10 min, and then returned to 15 °C seawater, most of the cells died (25% viability as shown in Figure 1A). Thus, aeration-cultured ‘*Bangia*’ sp. ESS1 has little tolerance to direct transfer from 15 °C to sub-zero temperature. By contrast, when aeration-cultured thalli were statically cultured at 15 °C for 1–6 weeks prior to freezing, cell viability was significantly higher, gradually increasing with the duration of static culture to 90% (Figure 1A). Viability was maintained for 1 week after freeze-and-thaw treatment (Appendix A). In addition, when similarly treated samples were directly transferred to 32 °C seawater, as a lethal heat-stress condition [36], after freezing, viability decreased depending on the duration of heat-stress exposure (Appendix A). Moreover, static culture of thalli at 15 °C for 2 weeks or 6 weeks accelerated the release of asexual spores after freeze-and-thaw treatment and subsequent 1 week-culture at 15 °C; this did not occur in statically cultured thalli without freezing (Figure 1B).

### 2.2. Unsaturation of Membrane Fatty Acids under Calm Stress Conditions

To examine whether the membrane fatty acid composition is modulated by calm conditions, we incubated aeration-cultured thalli at 15 °C in static culture for 1–6 weeks and analyzed membrane fatty acid compositions of samples harvested at every week. The relative amounts of saturated fatty acids and monoenes decreased compared to those in aeration-cultured thalli, whereas the relative amounts of polyenes gradually increased (Appendix A). The significance of decreases in saturated fatty acids and monoenes and increases in polyenes was clearly demonstrated by comparison of the fatty acid compositions among aeration-cultured thalli and samples that were statically cultured for 2 weeks or 6 weeks (Figure 2). Since the main saturated and unsaturated fatty acids were palmitic acid (16:0) and eicosapentaenoic acid (20:5 *n*-3), respectively (data not shown), the results of saturates and polyenes in Figure 2 roughly reflected the changes in contents of these fatty acids.

## 3. Discussion

We here demonstrated that calm stress promotes the acquisition of freezing tolerance and an increase in the unsaturation of membrane fatty acids, which enables survival upon exposure to −80 °C in ‘*Bangia*’ sp. ESS1 but does not induce thermotolerance. Remarkably, freezing tolerance was established with only 1 week of static culture. Thus, loss of water movement can mimic chilling as a priming stress that triggers the establishment of freezing tolerance, meaning that the acquisition of freezing tolerance is one strategy for the toleration of calm conditions. This finding is consistent with our previous results showing that an increased saturation of membrane fatty acids is required for the acquisition of heat-stress tolerance [37], which is reciprocally related to the decrease in saturation level by static culture shown in Figure 2 and suggests that membrane fluidization is critically involved in calm-stress responses. Although the positive contribution of calm conditions to gamete release has been demonstrated in green and brown algae [39,40,41], the finding that calm stress promotes the acquisition of freezing tolerance in algae is novel.

Our results also indicate that freeze-and-thaw treatment of statically cultured gametophytes enhances the release of asexual spores in ‘*Bangia*’ sp. ESS1, in which asexual reproduction is accelerated in a freezing-dependent manner. Thus, loss of water movement seems to increase sensitivity to freeze-and-thaw cycles for promotion of the asexual life cycle. We previously observed an enhancement of asexual reproduction by heat stress in this alga [36] and by hypo-osmotic, oxidative, and wounding stresses in the red alga *Pyropia yezoensis* [42,43,44]. Thus, we propose that environmental stress can trigger a transition from growth to reproductive phase in the life cycle of Bangiales. Consistently, gamete release by the depletion of dissolved inorganic carbon (DIC) was previously proposed under calm conditions in the brown alga *Fucus distichus* [39]; however, it is unknown whether reduced DIC content acts as a signal to promote the acquisition of freezing tolerance in ‘*Bangia*’ sp. ESS1. Therefore, studying the regulatory mechanisms underlying spore formation and release will further elucidate abiotic stress-inducible asexual reproduction and its relation to membrane fluidity in Bangiales.

Since hydrodynamic stress essentially occurs in the hydrosphere, an ability to acquire cross-tolerance to calm and freezing stresses seems to be unique to aquatic organisms. The acquisition of freezing tolerance by exposure to calm stress is a reasonable adaptation to the circumstances of a tide pool, where organisms may experience falling air temperatures and snow. However, little is known about the sensing of and signal transduction in response to loss of water movement in algae. Thus, the identification of factors that trigger and/or participate in freezing tolerance and membrane fatty acid unsaturation in response to loss of water movement could help us understand how calm-stress signaling is regulated and interacts with chilling signal transduction pathways in ‘*Bangia*’ sp. ESS1.

## 4. Materials and Methods

### 4.1. Algal Material and Stress Treatment

Thalli of the marine red seaweed ‘*Bangia*’ sp. ESS1 [35] were collected at Esashi, Hokkaido, Japan on 14 May 2010 [38], and a clean single thallus of unknown sex was aeration cultured at 15 °C and maintained as an experimental line. For static culture, 0.1 g (fresh weight) samples of thalli were cultured in dishes (Petri dish φ90 × 20 mm height) containing 50 mL of enriched SEALIFE (ESL) medium [35] at 15 °C under 60 μmol m^−2^ s^−1^ irradiation for 0 (control) to 6 weeks. Algal samples harvested every week were stored at −80 °C for 10 min and then cultured at 15 °C for 0 (just after thawing) and 1 week as a freezing-stress treatment or cultured at a lethal temperature of 32 °C for 0, 1, 3, 5, and 7 days as a heat-stress treatment. Samples harvested at every week of freezing treatment and all durations of heat-stress treatment were subjected to analyses of viability and membrane fatty acid composition as described below.

### 4.2. Viability Test and Observation of Asexual Spore Release

The viability of cells of statically cultured and freeze-and-thaw treated thalli was examined as described previously [36,37]. In brief, ‘*Bangia*’ sp. ESS1 thalli exposed to calm and freezing stresses as described above were stained daily with ESL medium containing 0.01% erythrosine (Wako Pure Chemical Industries, Osaka, Japan). After staining for 20 min at room temperature, thalli were gently rinsed with ESL medium to remove excess erythrosine and mounted on slides with ESL medium. Thalli were observed and photographed using an Olympus IX73 light microscope equipped with an Olympus DP22 camera. Cells stained by the dye were defined as dead cells, as indicated in Takahashi et al. [45]. Viability was calculated from the number of living and dead cells obtained using micrographs. The observation of asexual spore release from statically cultured and freeze-and-thaw treated thalli was performed microscopically as described above. The ratio of the number of asexual spore-releasing thalli to the total number of thalli was calculated.

### 4.3. Analysis of Membrane Fatty Acid Composition

Fresh samples of ‘*Bangia*’ sp. ESS1 were immersed in boiling water for 3 min to deactivate lipid hydrolytic enzymes and then freeze-dried and homogenized using a grinder. Lipids were extracted from 0.1 g powdered algal sample via the Bligh–Dyer method [46] with some modifications as described in Kishimoto et al. [37]. The preparation of fatty acid methyl esters based on Christie and Han [47] and its GC analysis using a Shimadzu GC-14A gas chromatograph (Shimadzu Corporation, Kyoto, Japan) with an Omegawax 320 column (30 m × 0.32 mm i.d., Supeleo, PA, USA) were performed as described previously [37].

### 4.4. Statistical Analysis

Values are indicated with SD from triplicate experiments. A one-way ANOVA followed by a Tukey–Kramer test was used for multiple comparisons, and significant differences were determined using a cutoff value of *p* < 0.05 as described in [48].

## 5. Conclusions

‘*Bangia*’ sp. ESS1 acquires freezing tolerance when it is exposed to calm stress, for which an increase in unsaturation levels of membrane fatty acids might be involved. Recent studies have indicated the involvement of membrane integrity based on lipid remodeling in freezing tolerance in terrestrial plants [26,49,50,51], although changes in membrane lipid compositions by transferring from aeration to static culture conditions were not analyzed in algae. Therefore, elucidation of the relationship between calm-stress signaling and lipid remodeling in membranes under null hydrodynamic stress conditions in ‘*Bangia*’ sp. ESS1 could provide insights into the unique characteristics that regulate the acquisition of freezing tolerance and asexual life cycle by calm stress in Bangiales.

## Figures and Tables

**Figure 1 plants-10-00465-f001:**
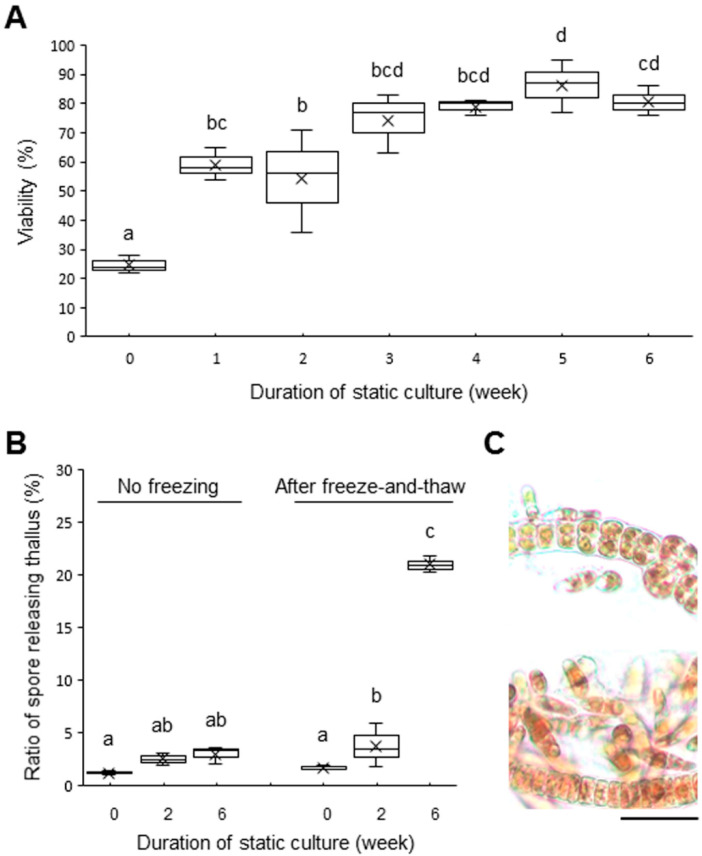
Effects of static culture on viability and release of asexual spores in ‘*Bangia*’ sp. ESS1. (**A**) Static culture-induced increase in viability after freeze-and-thaw. (**B**) Enhancement of release of asexual spores from statically cultured thalli after freeze-and-thaw. (**C**) Extensive release of asexual spores in thalli exposed to freeze-and-thaw treatment (lower) than non-frozen thalli (upper) after static culture for 6 weeks. Most released spores developed into small germlings. Scale bar: 50 μm. Letters on boxes denote significant differences from triplicate experiments defined by the Tukey–Kramer test (*p* < 0.05) in one-way ANOVA.

**Figure 2 plants-10-00465-f002:**
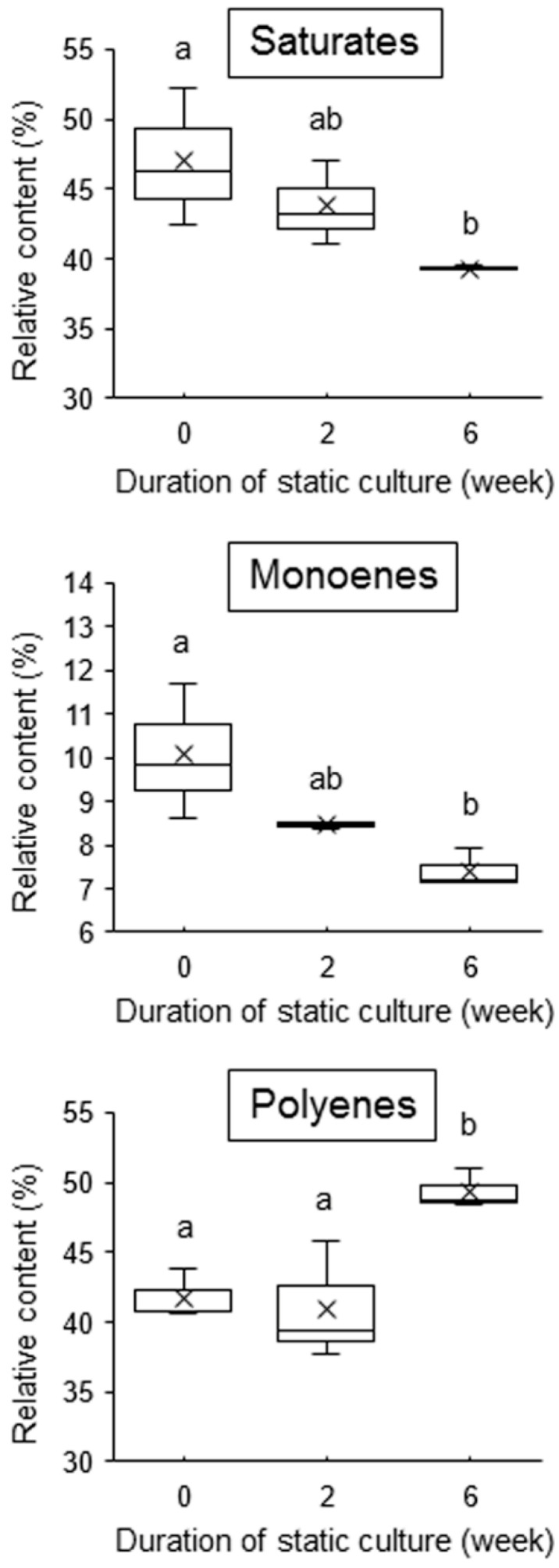
Effects of static culture on membrane fatty acid compositions in ‘*Bangia*’ sp. ESS1. Changes in the relative amounts of saturated fatty acids (Saturates), monounsaturated fatty acids (Monoenes), and polyunsaturated fatty acids (Polyenes) were analyzed in thalli statically cultured for 2 and 6 weeks in comparison with control samples (0) without static culture. Letters on boxes denote significant differences from triplicate experiments defined by the Tukey–Kramer test (*p* < 0.05) in one-way ANOVA.

## Data Availability

Data is contained within the article.

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
