# Peer review of "The Absence of Hydrodynamic Stress Promotes Acquisition of Freezing Tolerance and Freeze-Dependent Asexual Reproduction in the Red Alga ‘Bangia’ sp. ESS1"

_plants, 2021, doi:10.3390/plants10030465_

Round 1

Reviewer 1 Report

Dear Authors,

the presented manuscript deals with an interesting topic especially if we consider that in the Bangiales order there are very important economic genres such as Porphyra and Pyropia. In the manuscript it is not clear how water movement crossed with temperature changes could be involved in changes in the membrane structure, however the input the manuscript gives can be considered a good start for future studies. In my opinion the manuscript, although preliminary, could be published after some minor revisions.

Reviewer 2 Report

Introduction

  1. Land plants have interconnected signaling pathways than can be stimulated by multiple environmental conditions to activate tolerance to several stress factors, usually phytohormones signaling is involved in those interconnected responses. Does Bangiales’ omics and physiological studies have revealed some of those signaling protein elements being activated or upregulated?
  2. Lipid modifications are known to confer temperature tolerance in several lineages, what is known about glycosylation of lipids in Bangiales regarding cold tolerance?
  3. What is known about the oxygen levels in calm water compared to water in movement? Maybe the lack of oxygen is activating the tolerance mechanism instead the lack of movement. As well as desaturases will not be able to produce unsaturated lipids since they obtain electrons from molecular oxygen.

Results

  1. The transfer of the alga to -80° seems to be too extreme and far from natural freezing conditions. What was the reason this freezing temperature was chosen?
  2. What are the tolerance limits of Bangia regarding the lower temperature and the higher temperature in its natural environment? This is to have an idea of why 32° was chosen as the lethal heat-stress condition.

Discussion

  1. Interestingly polyenes were increased compared to control conditions. What were the parameters you measure to determinate the culture conditions were indeed stress conditions? Only the viability assay is provided for this communication, maybe you can include some supplementary figure about reactive oxygen species levels or antioxidant enzyme activation levels, if available. Could the increase in polyenes be the product of oxidative stress?
  2. What effects of chilling, other than membrane lipid composition, are comparable between calm-stress and freezing-stress physiological responses?
  3. I’m concerned the formation of spores was induced by lack of carbon dioxide in water since the calm-stress cultures did not have bubbling. Do you have information about photosynthetic activity under this calm-stress condition? since freezing is also known to decrease photosynthesis.

Materials and methods

  1. Line 160 says “satirized artificial seawater”, please correct.
  2. What is the composition of the artificial seawater, please include the information about the provider or the salts used to reconstitute the seawater?
  3. How many replicates of each culture were performed?
  4. Which were the main saturated and unsaturated lipids revealed by GC? Please include a supplementary table or briefly mention in the results section.

Round 2

Reviewer 2 Report

The autors provided the additional data asked and corrected the lines with observations.

Also kindly answered all the questions of this revision with enough supporting information.

The new version of the manuscript has improved and is suitable for publication.